# Therapy of genomic unstable solid tumours (WHO grade 3/4 ) in clinical stage III/IV using individualised neoantigen tumour peptides-INP trial (individualised neoantigen tumour peptides immunotherapy): study protocol for an open-label, non-randomised, prospective, single-arm trial

Ling Wang [1], Jiaxi Tang [2], Xia Chen,[3] Juan Zhao,[1] Wanyan Tang,[1] Bin Liao,[1] Weiqi Nian[1,4]

LW and JT contributed equally.

LW and JT are joint first authors.

For numbered affiliations see end of article.

**Correspondence to**
Professor Weiqi Nian;
nwqone@gmail.com

## ABSTRACT

**Introduction** Neoantigens derived from tumour somatic mutations are recognised as ideal vaccine targets. Tumour neoantigens have been studied in a wide range of tumours. Most of research on neoantigens has focused just on a unique tumour and a single mutated gene. Currently, a few studies have reported using a mixture of neoantigen peptides derived from multiple genetic mutation sites in the treatment of genomic unstable advanced solid malignancies. The trial aims to evaluate the safety and efficacy of individualised tumour neoantigen peptide mixtures in the treatment of genomic unstable advanced solid malignant tumours.

**Methods and analysis** This is a prospective, non-randomised, open, single-centre, single-arm, phase I trial. Patients with genomic unstable advanced solid malignancies are eligible for study participations. 20 patients will be included in the trial. Through the whole exome and transcriptome sequencing analysis of the fresh blood and tumour tissues of the enrolled patients, the 20 25-33aa antigen peptides with the highest mutation scores of the patients will be screened out, and the corresponding new antigen peptides will be synthesised and prepared. Patients will be treated with their own individualised neoantigen polypeptide combined with a polypeptide adjuvant (human granulocyte-macrophage colony-stimulating factor). The primary endpoint is safety indicators, including general and specific adverse events which will be monitored continuously. Secondary endpoints are progression-free survival, objective response rate, objective duration of remission, 1-year survival rate and overall survival.

**Ethics and dissemination** This study has received approval from the Ethics Committee of Chongqing

## STRENGTHS AND LIMITATIONS OF THIS STUDY

⇒ In this study, the design and synthesise of individualised neoantigen tumour peptides will be completed by a professional team to ensure the quality of tumour neoantigen.
⇒ The study has clearly defined the entry and exclusion criteria for inclusion in this clinical trial.
⇒ Rigorous adverse events observation and follow-up give opportunities to evaluate safety of patients, allowing for early trial intervention if necessary.
⇒ This trial only includes subjects with a high mutation burden, and its suitability for subjects with a low mutation burden needs further research.
⇒ Since this study is a phase I trial, the sample size is limited, so future clinical studies with a larger sample size are necessary.

University Cancer Hospital on 21 November 2019 (207/2019). The findings of this trial will be disseminated through national and international presentations and peer-reviewed publications.

**Trial registration number** ChiCTR1900025364.

## INTRODUCTION

Malignant tumours are a serious threat to human life and health and have become the second leading cause of human death.[1] At present, many tumours cannot be cured by conventional methods such as surgery, radiotherapy and chemotherapy. With the rapid development of biotechnology and the

in-depth study of the molecular mechanisms of tumour occurrence and development, biotherapy has become the fourth mode of comprehensive tumour treatment and has received more and more attention.[2–4]

Tumour biotherapy is a new therapy that applies modern biotechnology and its products for tumour prevention and treatment. It achieves antitumour effects by mobilising the host's natural defence mechanism or giving natural (or genetic engineering) highly targeted substances. It mainly includes somatic cell therapy, cytokine therapy, tumour peptides therapy, molecular targeted therapy, radioimmune targeted therapy, tumour gene therapy and biological chemotherapy.[3–9] Among them, tumour neoantigen study is one of the hotspots in domestic and foreign research in recent years.[10] Individualised neoantigen tumour immunotherapy has brought inspiring therapeutic effects. Several studies have shown that tumour neoantigen therapy can produce miraculous effects, inhibiting tumour progression and even achieving complete tumour regression.[11–13]

Tumour neoantigens are a series of immunogenic substance derived from tumour-specific mutations, instead of from the normal human genome,[14–17] which presented on the surface of tumour cells by major histocompatibility complex (MHC). The new antigen can be transferred into the patient in the form of synthetic long peptide, RNA, activated dendritic cell and DNA.[18–21] Neoantigens are not present in normal cells, so the immune system can recognise them as 'non-self' antigens.[22] The principle is to eliminate or control tumours by activating the patient's own immune system.[20] In the body, neoantigen epitopes can bind to MHC-I to activate CD8 +T lymphocytes, which can differentiate CD8 +T lymphocytes into cytotoxic T lymphocytes (CTL) and exert cytotoxic effects. Also, neoantigen epitopes can combine with MHC-II to activate CD4 +T lymphocytes to differentiate into helper T cells, such as Th1 and Th2 cells, which interact with macrophages and B lymphocytes, respectively.[23 24]

Tumour neoantigens have been studied in a wide range of tumours, including lung, breast, glioma, liver, kidney, ovarian, pseudomyxoma peritonei and pancreatic cancer, etc.[5 12 18 25–28] Most of research on neoantigens has focused just on a unique tumour and a single mutated gene.[5 11 29] The safety and efficacy of neoantigen peptide mixtures derived from multiple gene mutation sites in the treatment of advanced solid malignancies with genomic instability are still unclear.

The trial aims to evaluate the safety and efficacy of individualised tumour neoantigen peptide mixtures in the treatment of genomic unstable advanced solid malignant tumours.

## METHODS
### Study design overview
This is an open-label, non-randomised, prospective, single-arm trial, which will be conducted at phase I Trial Ward of Tertiary Class A Cancer Hospital.

### Objectives
The trial aims to evaluate the safety and efficacy of individualised tumour neoantigen peptides mixtures in the treatment of genomic unstable advanced solid malignant tumours.

### Endpoints
The primary endpoint is to observe and record all adverse events (AEs) and the incidence of serious AEs (SAE) (grade 3 or above AE) associated with the study drug within 24 weeks after the administration of the individualised neoantigen tumour polypeptide (Refer to National Cancer Institute Common Terminology Criteria for Adverse Events (NCI) version 4.03). After that, the subjects will continue to be followed at the preset follow-up points for safety. Secondary endpoints are to evaluate the progression-free survival (PFS), objective response rate (ORR), objective duration of remission (DOR), 1-year survival rate and overall survival (based on Enhanced CT/MRI). The efficiency of neoantigen peptides in inducing specific antitumour immunity is assessed from peripheral blood samples (based on IFN-γ, CD4 +T cells, CD8 +T cells determined by flow cytometry and enzyme-linked immuneospot (ELISPOT) technique. In addition, the efficacy evaluation also includes the detection of circulating tumour DNA (ctDNA) and the quality of life of cancer patients.

### Additional scientific programme
#### Genome sequencing
Whole exome and transcriptome sequencing will be tested on the patient's biopsy or surgically removed tumour tissue, and whole exome sequencing will also be performed on the patient's blood sample. This gene sequencing can obtain comprehensive genomic information such as the mutation status of all genes, human leucocyte antigen (HLA) genotypes and expression levels of mutated genes in the patients, which can be used to confirm patient's entry criteria and synthesise neoantigen tumour peptides.

### Design and synthesise of neoantigen tumour peptides
The main body of the neoantigen design process in this study refers to the proposal published in Nature in 2018 by Neon Therapeutics and Harvard Medical School.[28] On this basis, the unique analysis technology of Beijing Xinkangyuan Biotechnology will be applied to further ensure the quality of tumour neoantigen peptides. Beijing Neoantigen Biotechnology owns a series of software copyrights such as NeoRQC, NeoAlign, NeoHLAType, NeoCVC, NeoVQC, NeoANN, NeoGEP, NeoDsign and other softwares which are under the NeoOne platform specially designed for neoantigen design and synthesis. They will be used for gene mutation analysis, gene expression analysis, HLA typing, neoantigen design and multidirectional quality control to ensure that the neoantigen design in this study meets the world's leading clinical requirements. The whole design process is shown in figure 1.

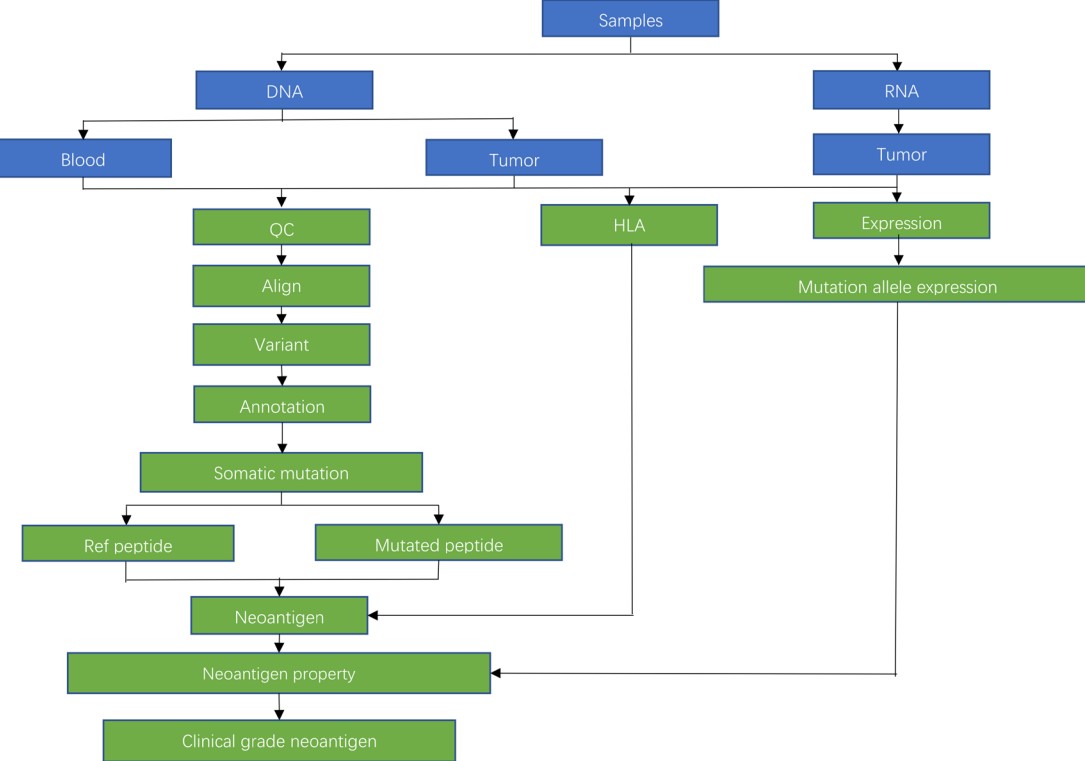

**Figure 1** Design flow chart of neoantigens. HLA, human leucocyte antigen; QC, quality control.

By analysing the whole exome and transcriptome sequencing-specific somatic mutations of fresh blood samples and tumour tissues of the enrolled patients, 20 antigen peptides sequences of 25–31aa with the highest mutation scores will be screened out. According to the above-mentioned designed individualised neoantigen polypeptide sequence, the Fmoc solid-phase method will be used to synthesise the polypeptide. Ninhydrin colour reaction will be used for quality control. The crude peptide will be purified by high-performance liquid chromatography (HPLC) method, then the quality of peptide will be checked by mass spectrometer (MS) and HPLC. Qualified peptides should meet two criteria: correct MS and HPLC purity above 95%. Individualised neoantigen tumour peptides for the treatment of genomic unstable solid tumours use 20 peptide mixtures. According to the Good Manufacturing Practice standard of the raw material drug, the peptides will be separated and frozen for subsequent trials.

### Inclusion criteria
1. Gender is not limited.
2. Patients range in age from 18 to 70 years old.
3. Patients who have progressed after comprehensive treatment and diagnosed with grade 3/4 (WHO) genomic unstable (tumour mutational burden >10 or microsatellite instability-high (MSI-H)) solid malignant tumours, and lack further effective treatment, or the guidelines at the time of diagnosis of the patient have no corresponding recommended treatment plan are eligible.

4. There is at least one lesion that can be evaluated.

### Exclusion criteria
1. Patients who have a history of hypersensitivity to the testing drug ingredients.
2. Patients with a previous history of gene therapy or any history of antitumour therapy in the last 4 weeks.
3. Female patients who are currently pregnant, suspected of being pregnant, or expect to become pregnant during the study period.
4. Patients who are judged to be inappropriate as study subjects.
5. Patients who lack sufficient tumour tissues and peripheral blood samples for gene sequencing.

### Termination criteria
Patients can request withdrawal from the clinical trial at any time. If it becomes difficult to administrate the testing drug to the subject due to the following reasons, the responsible physician or the sharing physician will decide to discontinue the drug administration.
1. Any time it is found that the patient does not meet the inclusion criteria or meets the exclusion criteria.
2. Uncontrollable T cell proliferation.
3. SAEs, regardless of whether they are related to treatment.
4. Tumour progression.

### Study intervention and timeline of the study
Individualised tumour peptide mixtures of 20 antigenic peptides will be injected subcutaneously for therapeutic

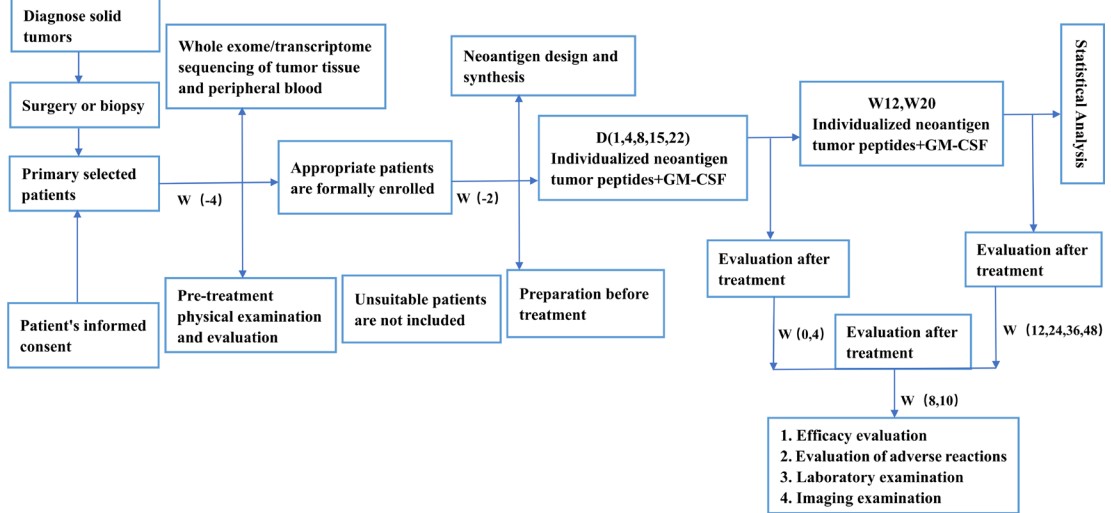

**Figure 2** The study flow chart. GM-CSF, granulocyte-macrophage colony-stimulating factor.

intervention in patients. The benchmark dose is 0.3 mg for each antigen peptide, and the single injection dose is 0.3 mg ×20=6.0 mg. Granulocyte-macrophage colony stimulating factor (GM-CSF) will be selected as a polypeptide adjuvant, with the dosage of 0.5 µg/kg/dose.

The medicine combine individualised tumour antigen polypeptides and GM-CSF will be injected subcutaneously at the lower edge of the upper arm deltoid muscle. The immunotherapy injection cycle is divided into two phases: initial immunisation injection and booster immunisation injection, which are completed in days 1, 4, 8, 15, 22 and weeks 12, 20. During the initial injection of individualised neoantigen tumour peptides, AEs will be recorded on the day of injection. During booster immunisation, AEs will be recorded for three consecutive days after each injection. The efficacy of immunotherapy will be evaluated at 4, 8, 12, 24, 36 and 48 weeks. The timeline is displayed in figure 2.

The individual follow-up for each participant is 12 months. Patients may enter in a succeeding phase (extended follow-up) according to the preset standards. The trial started in Q3/2019 and will last about 5 years.

### Assessment of therapeutic safety and efficacy

After treatment with individual neoantigen tumour polypeptides, all drug-related AEs occurring within 24 weeks will be observed and recorded. Adverse reactions that require close attention during the trial include cytokine storm-related acute inflammatory response syndrome, allergic reactions, neurotoxicity and systemic infections.

PFS, ORR and DOR will be evaluated by CT/MRI according to the NCI criteria.[30] Quality of life will be assessed using the Quality-of-Life Questionnaire of the European Organisation for Research and Treatment of Cancer 30.[31] In addition, 2 mL peripheral blood samples will be collected and IFN-γ, CD4+, CD8 +antigen signals in peripheral blood will be detected by flow cytometry to evaluate the effectiveness of tumour neoantigen polypeptide in inducing specific anti-tumour immunity. The

changes of IFN-γ cells, CD4 +T cells, and CD8 +T cells in the peripheral blood mononuclear cell before and after immunotherapy will be analysed with ELISPOT technology. The efficacy evaluation also includes ctDNA detection before and after neoantigen immunotherapy.

The observed indicators also include physical examination, vital signs, Eastern Cooperative Oncology Group score, weight, blood routine, urine routine, liver and kidney function, coagulation function and tumour markers. At weeks 8, 12, 24, 36 and 48 of the study periods, patients' target lesions will be scanned using enhanced CT or MRI to evaluate the therapeutic effect. The intervention and evaluation schedule are shown in table 1.

### Sample size

To the best of our knowledge, there are no objective data on the efficacy of individualised neoantigens in advanced tumours. This study will not stipulate a sample size a priori. The sample size is not therefore determined by power calculations. The purpose of the trial is to evaluate the safety of individualised neoantigens, similar to the phase I clinical trials of drugs, so our sample size is determined to be 20 patients.

### Statistical considerations

First, we need to list the number of selected and completed cases, confirm three data sets: full analysis set, per-protocol set, safety data set and then describe demographic data and other baseline characteristic values.

Descriptive data analysis will be used to evaluate the experimental data. The percentage of subjects with medication compliance greater than 80%, and the percentage of subjects with combination medication will be calculated. AEs will be coded according to International Conference on Harmonisation (ICH) Dictionary of International Medical Terms (MedDRA). List AE, SAE, major AE, AE leading to disengagement and calculate the incidence. Data from patients who received at least five sessions of individualised tumour neoantigen polypeptide

**Table 1** Intervention and assessment schedule for the INP trial

| Execute project | W-4–W0 Screening period | W1 | W2 | W3 | W4 | W5 | W6 | W8 | W10 | W12 | W20 | W24 | W36 | W48 |
|---|---|---|---|---|---|---|---|---|---|---|---|---|---|---|
| Sequencing/neoantigen preparation | √ | | | | | | | | | | | | | |
| Neoantigen immunisation | | √√ | √ | √ | √ | | | | | √ | √ | | | |
| Physical examination | √ | √ | √ | √ | √ | √ | √ | √ | √ | √ | √ | √ | √ | √ |
| Laboratory examination | √* | | | | √ | | | √ | | √ | √ | √ | √ | √ |
| CT/MRI | √* | | | | | | | √ | | √ | | √ | √ | √ |
| ctDNA detection | √* | | | | √ | | | √ | | √ | | √ | √ | √ |
| IFN-γ, CD4 +T cells, CD8 +T cells | √* | | | | √ | | | | | | √ | | | √ |
| AEs (CTCAE) | √* | √ | √ | √ | √ | √ | √ | √ | √ | √ | √ | √ | √ | √ |
| QLQ (EORTC QLQ-C30) | √* | | | | √ | | | √ | | √ | | √ | | √ |
| Survival follow-up | | √ | √ | √ | √ | √ | √ | √ | √ | √ | √ | √ | √ | √ |

√√ Twice a week.
*Need to screen out qualified neoantigens before relevant testing.
AE, adverse event; CTCAE, Common Terminology Criteria for Adverse Events; ctDNA, circulating tumour DNA; EORTC QLQ-C30, European Organisation for Research and Treatment of Cancer Quality of Life Questionnaire; INP, Individualized neoantigen tumor peptides.

immunotherapy will be included in the clinical outcome and safety assessment. GraphPad Prism V.8 will be used to plot the ctDNA dynamic curves and immune response curves.

## Patient and public involvement

Patients and the public will understand the design and benefits of our trial through consulting with patient advisors and reviewing recruitment information. The patients do not participate in the design of the experiment. The results of the experiment will be kept confidential to the patients.

## Ethics

The procedures set out in this trial protocol are designed to ensure that all persons involved in the trial abide by the ICH harmonised tripartite guideline on Good Clinical Practice (ICH-GCP) and the ethical principles described in the applicable version of the Declaration of Helsinki. The trial will be carried out in keeping with local legal and regulatory requirements. The regulations of the GCP regulations will be respected. Before the start of the trial, all documents have been submitted to the independent ethics committee. The study protocol has been approved by local ethics committee on 21 November 2019 (Protocol version 1.2 (27 September 2019)).

## Data quality assurance

The histological diagnosis will be verified by one of the experienced pathologists. Tumour grading and evaluation of post-treatment curative effect should be decided by public consultation with experienced clinicians and radiologists.

To ensure the quality of the data, we will follow the standard operating procedures of the Clinical Trial Quality Management Practice Guidelines (CPMP/ICH/135/95) throughout the trial. During the clinical trial process, the supervisor will send the case report form (CRF) to the data management unit in real time, and two data administrators will perform independent double entry and double check. Inconsistent results will be checked and corrected item by item according to CRF.

## Trial status

The trial started in September 2019 and is currently recruiting. The length of clinical phase is about 60 months. (The planned end of the study is the end of 2024). The first patient was treated in August 2020.

## DISCUSSION

Tumour neoantigens are derived from somatic cell mutations, which are present in various tumours, and there is a strong heterogeneity among them.[32] Cancer neoantigen therapy was first reported in 1979, and this study showed that successfully treated lymphoma patients responded less well to tumour neoantigen therapy.[33] Further studies also suggest that patients with low mutation burden may be limited in their use of neoantigen therapy due to the restricted number of immunogenic epitopes.[34] In view of this, our study included patients with a high mutation load.

Current studies suggest that T-cell-based immune responses are involved in the process of tumour neoantigen immunotherapy.[10 28 35 36] The most striking feature of T cells is their memorability, which reflects the

long-term and sustained clinical results achieved by tumour immunotherapy. Wedén *et al* followed up for 10 years in patients with pancreatic cancer and found that survival was 20% (4/20) in patients treated with a k-RAS mutant neoantigen vaccine compared with 0% (0/87) in the control group.[37] The study by Schreiber's group highlights the critical role of MHC-II in the presentation of neoantigens. MHC-II can induce the polarisation of Th1, promote the activation of CTL, and increase the antitumour ability of CTL.[38 39]

The personalised neoantigen vaccine based on high-throughput sequencing has the advantages of accurately identifying and killing tumour cells, which is expected to be an important breakthrough for vaccines towards clinical practice. As a specific target of immunotherapy, the antitumour effect of tumour neoantigens can only target tumour cells rather than normal tissues.[40] The identification of neoantigens and the development of neoantigens vaccine may provide more options for clinical immunotherapy. It may become one of the effective ways to treat malignant tumours in the future. This trial will be able to evaluate the safety and efficacy of neoantigen therapy in genomic unstable advanced solid malignant cancer patients and reveal the potential immune effects of individualised neoantigen tumour peptides mixtures in cancer patients.

## ETHICS AND DISSEMINATION
### Research ethics approval
This study has received approval from the Ethics Committee of Chongqing University Cancer Hospital on 21 November 2019 (207/2019). The study protocol, informed consent form and other submitted documents were reviewed and approved.

### Confidentiality
During the clinical trial, subjects will be identified solely by means of their individual identification code. Trial data will be stored and encrypted in accordance with local data protection laws.

### Dissemination policy
Regardless of the results of the research, the final data will be publicly released. After the data collection is completed (within 12 months of the end of the study), a report with the results of the research will be submitted in an appropriate open-access, peer-reviewed medical journal.

**Author affiliations**
$^1$Department of Phase I Clinical Trial Ward, Chongqing University Cancer Hospital, Chongqing, China
$^2$Department of Anesthesiology, Chongqing University Cancer Hospital, Chongqing, China
$^3$Department of Clinical Trial Center, Chongqing University Cancer Hospital, Chongqing, China
$^4$Department of Oncology, Chongqing Hospital of Traditional Chinese Medicine, Chongqing, China

**Acknowledgements** This work was supported by Beijing Neoantigen Biotechnology in terms of technology. We acknowledge financial support from Special Project for Performance Incentive and Guidance of Scientific Research Institutions of Chongqing, China grant to LW and XC, Beijing Municipal Science and Technology Commission to YMZ, and Natural Science Foundation and Science and Technology Bureau joint Health Bureau of Chongqing, China grant to JXT. We also thank the patient advisers for their efforts throughout the study.

**Contributors** LW, JT, XC, WT, JZ and BL developed and planned this trial, and WN is the principal investigator of the study. LW performed basic research and wrote the manuscript. JT is responsible for statistical planning and statistical analysis. All authors read and approved the final manuscript.

**Funding** The trial is sponsored by Special Project for Performance Incentive and Guidance of Scientific Research Institutions of Chongqing, China (cstc2019jxjl130029), Special Project for Performance Incentive and Guidance of Scientific Research Institutions of Chongqing, China (cstc2018jxjl130057), Science and Technology Bureau joint Health Bureau, Chongqing, China (2021MSXM177), Beijing Municipal Science and Technology Commission (Z191100007619018), and Natural Science Foundation of Chongqing, China (cstc2019jcyj-msxmX0623).

**Competing interests** None declared.

**Patient and public involvement** Patients and/or the public were not involved in the design, or conduct, or reporting, or dissemination plans of this research.

**Patient consent for publication** Not applicable.

**Provenance and peer review** Not commissioned; externally peer reviewed.

**ORCID iDs**
Ling Wang http://orcid.org/0000-0002-3329-6771
Jiaxi Tang http://orcid.org/0000-0001-5023-9173

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
