## [Reviewer comments · BMJ Open]

ARTICLE DETAILS

TITLE (PROVISIONAL)	Therapy of genomic unstable solid tumors (WHO grade 3/4) in clinical stage III/IV using individualized neoantigen tumor peptides-INP Trial (Individualized neoantigen tumor peptides Immunotherapy):study protocol for an open-label, non- randomized, prospective, single-arm trial
AUTHORS	Wang, Ling; Tang, jiaxi; Chen, Xia; Zhao, Juan; Tang, Wanyan; Liao, Bin; Nian, Weiqi

VERSION 1 – REVIEW

REVIEWER	Yong Qiong Deng The Affiliated Hospital of Southwest Medical University
REVIEW RETURNED	22-Feb-2022

GENERAL COMMENTS	This manuscript aimed to show the process about design and synthesize of individualized neoantigen tumor peptides and to evaluate the safety and efficacy of individualized neoantigen tumor peptides in the treatment of genomic unstable solid malignant tumors. The motivation behind the problem investigated in this manuscript is interesting and meaningful, the manuscript has little expository and technical shortcomings, as detailed in the follows. 1.In the introduction section, the authors didn't fully elaborate the shortcomings of this research field, the significance, purpose and innovation of this research.2.Author should to clarify whether the sample size is 20 cases or 20 cases at most.3.Some tense errors need to be checked carefully. Such as in Lines 39-40 on page 2, "were"should be changed to "will be". I suggest you go over your manuscript carefully to see if there are any similar errors and then revise it.4.Lines 43-44 on page 2, lines 60 on page 3, Safety indicators are not explained clearly enough, acronyms are suggested to be defined when they are first mentioned, suggest to be changed to:" The primary endpoint was Safety indicators, including general and specific adverse events (AE) which be monitored continuously."5.In the introduction section, on the introduction of biotherapy, a literature review is recommended, inserting relevant references. For the introduction of tumor neoantigen research, it is recommended to insert the latest research in the past 5 years.6.Regarding acronyms, there are some irregularities in the use of acronyms. It is recommended to use acronyms when the words are mentioned for the first time. Please carefully check the irregular abbreviations in the manuscript and make corrections.
--

REVIEWER	Ji-Yuan Zhou Guangzhou Medical University Second Affiliated Hospital
REVIEW RETURNED	23-Feb-2022

GENERAL COMMENTS	In this protocol, the authors aimed to show design and synthesize individualized tumor peptides, and evaluate their safety and efficacy in the treatment of genomic unstable solid malignant tumors. There are no major problems with the description as it provides a sufficient amount of information in the content. It is a well-presented manuscript that would be of very interest to the potential reader of the journal and community in general. However, I would like the authors to consider the following points to improve its form before acceptance for publication. My minor comments are as following:  1. One major concern is the choice of statements with no linked references, such as written content in the Introduction section and the section of Miraculous effects in the treatment of malignant tumors using neoantigens. Some new-added references published in the last five years are suggested. 2. All the abbreviated format should be paid attention to. Please check the corresponding abbreviation. For example: what's the meaning of MS and WHO? Which types of MHC-II and MHCII is correct? 3. Introduction section: A transitional description is recommended to be added between the following two sentences to make it easier for potential readers to understand: "Individualized neoantigen vaccine developed on the basis of sequencing is expected to be an important breakthrough of precision therapy for tumors." "The trial aims to design and synthesize individualized tumor peptides, and evaluate the safety and efficacy of individualized tumor peptide therapy in the treatment of genomic unstable advanced solid malignant tumors." 4. Overall language quality is good. But some sentence polishing is required. Some are listed: "Agency and is endorsed by the National Health Service." "patient has at least one lesion can be assessed." 5. Also, some punctuation and uppercase and lowercase letters need to be taken care of, such as the writing of the section title "sample size". The sentence ".....potential Immune effect of individual neoantigenic tumor peptides....." Please correct it. 6. Figure 2 is a bit blurry compared to Figure 1, and the font sizes in the two figures are inconsistent. Please improve figure 2.
--

VERSION 1 – AUTHOR RESPONSE

Reviewer #1:

General comments:

This manuscript aimed to show the process about design and synthesize of individualized neoantigen tumor peptides and to evaluate the safety and efficacy of individualized neoantigen tumor peptides in the treatment of genomic unstable solid malignant tumors. The motivation behind the problem

investigated in this manuscript is interesting and meaningful, the manuscript has little expository and technical shortcomings, as detailed in the follows.

1. In the introduction section, the authors didn't fully elaborate the shortcomings of this research field, the significance, purpose and innovation of this research.
2. Author should to clarify whether the sample size is 20 cases or 20 cases at most.
3. Some tense errors need to be checked carefully. Such as in Lines 39-40 on page 2, "were" should be changed to "will be". I suggest you go over your manuscript carefully to see if there are any similar errors and then revise it.
4. Lines 43-44 on page 2, lines 60 on page 3, Safety indicators are not explained clearly enough, acronyms are suggested to be defined when they are first mentioned, suggest to be changed to: "The primary endpoint was Safety indicators, including general and specific adverse events (AE) which be monitored continuously."
5. In the introduction section, on the introduction of biotherapy, a literature review is recommended, inserting relevant references. For the introduction of tumor neoantigen research, it is recommended to insert the latest research in the past 5 years.
6. Regarding acronyms, there are some irregularities in the use of acronyms. It is recommended to use acronyms when the words are mentioned for the first time. Please carefully check the irregular abbreviations in the manuscript and make corrections.

Answer: Thank you for your comments on the paper. Thanks for your affirmation of this article. We have systematically reviewed the latest literatures on tumor neoantigens, and revised the introduction section. I would also like to thank you for your valuable comments on the grammar and writing of the article, which have greatly contributed to improve the quality of the article.

Specific Comments:

1. In the introduction section, the authors didn't fully elaborate the shortcomings of this research field, the significance, purpose and innovation of this research.

Answer: Thank you for the comments on the paper. We have revised the manuscript based on the reviewers' suggestions. (Page 5 line 9-18).

2. Author should to clarify whether the sample size is 20 cases or 20 cases at most.

Answer: Thank you for your questions and reminders. Our study's sample size is 20. We will change the contents which are not clear in the manuscript. (Page 2 line 6).

3. Some tense errors need to be checked carefully. Such as in Lines 39-40 on page 2, "were" should be changed to "will be". I suggest you go over your manuscript carefully to see if there are any similar errors and then revise it.

Answer: Thank you for your pertinent comments, which are of great help in improving the quality of our manuscript. (Page 2 line 10, 11,15,; page 7 line 18; page 8 line 1,9,11,12,14,19;page 9 line 1,17; page10 line11,14,21,22; page11 line 2,10,14; page12 line1,8; page15 line17,18).

4. Lines 43-44 on page 2, lines 60 on page 3, Safety indicators are not explained clearly enough, acronyms are suggested to be defined when they are first mentioned, suggest to be changed to:" The primary endpoint was Safety indicators, including general and specific adverse events (AE) which be monitored continuously."

Answer: Thank you for your comments and manuscript revision suggestions. We have revised the manuscript according to the comments of reviewer. In addition, we have carefully checked the misuse of acronyms in the manuscript and revised them. (Page 2 line 14-16; page3 line9; page6 line 16-17; page 8 line 14; page 9 line 3-5).

5. In the introduction section, on the introduction of biotherapy, a literature review is recommended, inserting relevant references. For the introduction of tumor neoantigen research, it is recommended to insert the latest research in the past 5 years.

Answer: Thank you for your comments. We have supplemented studies on tumor neoantigens and biotherapy, and cited references from the last 5 years. (Page 4 line 6,13; Page 5 line 12-16).

6. Regarding acronyms, there are some irregularities in the use of acronyms. It is recommended to use acronyms when the words are mentioned for the first time. Please carefully check the irregular abbreviations in the manuscript and make corrections.

Answer: Thank you for your suggestions and reminders, which are of great help to improve the language quality and readability of the article. We have carefully checked the misuse of acronyms in the manuscript and revised them. (page3 line9; page6 line 16-17; page 8 line 14; page 9 line 3-5).

Reviewer #2:

General comments:

In this protocol, the authors aimed to show design and synthesize individualized tumor peptides, and evaluate their safety and efficacy in the treatment of genomic unstable solid malignant tumors. There are no major problems with the description as it provides a sufficient amount of information in the content. It is a well-presented manuscript that would be of very interest to the potential reader of the journal and community in general. However, I would like the authors to consider the following points to improve its form before acceptance for publication. My minor comments are as following:

1. One major concern is the choice of statements with no linked references, such as written content in the Introduction section and the section of Miraculous effects in the treatment of malignant tumors using neoantigens. Some new-added references published in the last five years are suggested.
2. All the abbreviated format should be paid attention to. Please check the corresponding abbreviation. For example: what's the meaning of MS and WHO? Which types of MHC-II and MHCII is correct?
3. Introduction section: A transitional description is recommended to be added between the following two sentences to make it easier for potential readers to understand: "Individualized neoantigen vaccine developed on the basis of sequencing is expected to be an important breakthrough of precision therapy for tumors." "The trial aims to design and synthesize individualized tumor peptides, and evaluate the safety and efficacy of individualized tumor peptide therapy in the treatment of genomic unstable advanced solid malignant tumors."

4. Overall language quality is good. But some sentence polishing is required. Some are listed: "Agency and is endorsed by the National Health Service." "patient has at least one lesion can be assessed."
5. Also, some punctuation and uppercase and lowercase letters need to be taken care of, such as the writing of the section title "sample size". The sentence ".....potential Immune effect of individual neoantigenic tumor peptides....." Please correct it.
6. Figure 2 is a bit blurry compared to Figure 1, and the font sizes in the two figures are inconsistent. Please improve figure 2.

Answer: Thank you for your comments on our paper. Thanks for your affirmation of this article. We have systematically reviewed the latest literatures on tumor neoantigens, and revised the introduction section. I would also like to thank you for your valuable comments on the grammar and writing of the article, which have greatly contributed to improve the quality of the article. We have redrawn the illustrations in the article and unified the font size of the figures.

Specific Comments:

1. One major concern is the choice of statements with no linked references, such as written content in the Introduction section and the section of Miraculous effects in the treatment of malignant tumors using neoantigens. Some new-added references published in the last five years are suggested.

Answer: Thank you for your comments on our paper. We have inserted references from the last 5 years in the article where appropriate. (Page 4 line 3,10).

2. All the abbreviated format should be paid attention to. Please check the corresponding abbreviation. For example: what's the meaning of MS and WHO? Which types of MHC-II and MHCII is correct?

Answer: Thank you for your comments and reminders, which are of great help to improve the language quality and readability of the article. The usage of MHC-II and MHC-I is the right type. We have carefully checked the irregular phrasal expressions in the manuscript and revised them. (page3 line9; page5 line3,6; page6 line 16-17; page 8 line 14; page 9 line 3-5).

3. Introduction section: A transitional description is recommended to be added between the following two sentences to make it easier for potential readers to understand: “Individualized neoantigen vaccine developed on the basis of sequencing is expected to be an important breakthrough of precision therapy for tumors.” “The trial aims to design and synthesize individualized tumor peptides, and evaluate the safety and efficacy of individualized tumor peptide therapy in the treatment of genomic unstable advanced solid malignant tumors.”

Answer: Thank you for your comments. Considering that the introduction of the abstract part in our original manuscript is too long, and does not fully elaborate the shortcomings of this research field, the significance, purpose and innovation of our research. We have simplified and revised the introduction of the abstract part to improve the logic and readability of our article. (Page 1 line 17-23; page 2 line 1-3).

4. Overall language quality is good. But some sentence polishing is required. Some are listed: “Agency and is endorsed by the National Health Service.” “patient has at least one lesion can be assessed.”

Answer: Thank you for your comments and valuable advice. We have improved the writing problem. (Page 2 line 21; page 9 line 8).

5. Also, some punctuation and uppercase and lowercase letters need to be taken care of, such as the writing of the section title “sample size”. The sentence “.....potential Immune effect of individual neoantigenic tumor peptides.....”Please correct it.

Answer: Thank you for your comments and careful and responsible review. We have corrected these incorrect writing. (page 14 line 1; Page 18 line 3).

6. Figure 2 is a bit blurry compared to Figure 1, and the font sizes in the two figures are inconsistent. Please improve figure 2.

Answer: Thank you for your suggestions. We have redrawn the illustrations in the article and unified the font size of the figures. (See Fig.1 and Fig. 2)